

# A game theoretic analysis of research data sharing

Tessa E. Pronk[1], Paulien H. Wiersma[1], Anne van Weerden[1] and
Feike Schieving[2]

[1] Utrecht University Library, Utrecht University, Utrecht, The Netherlands
[2] Ecology and Biodiversity, Utrecht University, Utrecht, The Netherlands

## ABSTRACT

While reusing research data has evident benefits for the scientific community as a whole, decisions to archive and share these data are primarily made by individual researchers. In this paper we analyse, within a game theoretical framework, how sharing and reuse of research data affect individuals who share or do not share their datasets. We construct a model in which there is a cost associated with sharing datasets whereas reusing such sets implies a benefit. In our calculations, conflicting interests appear for researchers. Individual researchers are *always* better off not sharing and omitting the sharing cost, at the same time both sharing and not sharing researchers are better off if (almost) all researchers share. Namely, the more researchers share, the more benefit can be gained by the reuse of those datasets. We simulated several policy measures to increase benefits for researchers sharing or reusing datasets. Results point out that, although policies should be able to increase the rate of sharing researchers, and increased discoverability and dataset quality could partly compensate for costs, a better measure would be to directly lower the cost for sharing, or even turn it into a (citation-) benefit. Making data available would in that case become the most profitable, and therefore stable, strategy. This means researchers would willingly make their datasets available, and arguably in the best possible way to enable reuse.

## INTRODUCTION

While sharing datasets has group benefits for the scientific community and society as a whole, decisions to archive datasets are made by individual researchers. It is less obvious that the benefits of sharing outweigh the costs for all individuals (*Tenopir et al., 2011*; *Roche et al., 2014*). Many researchers are reluctant to share their dataset publicly because of real or perceived individual costs (*Pitt & Tang, 2013*). This probably explains why sharing datasets is no daily practice (*Roche et al., 2014*), especially when compared to sharing knowledge and information in the form of a scientific paper. Costs to individual researchers include time investment, money, the chance of being scooped by others on any future publications on the dataset, a chance that results from published papers will be over-scrutinized, misinterpretation of data resulting in faulty conclusions (*Atici et al., 2013*), misuse (*Bezuidenhout, 2013*), and possible infringement of the privacy of

Corresponding author
Tessa E. Pronk, T.E.Pronk@uu.nl

test subjects (*Antman, 2014*). Also, datasets are perceived as intellectual property and researchers simply do not want others to benefit from it (*Vickers, 2011*).

In contrast, the act of sharing research data could have advantageous consequences. Scientific outreach might be extended into other than the original research areas (*Chao, 2011*), and researchers' reputations could grow by the publicity of good sharing practices, possibly initiating new collaborations. In genetics (*Botstein, 2010*; *Piwowar & Vision, 2013*) it was calculated that papers with open data were cited more than studies without the data available. This citation advantage was also found in other disciplines like astronomy (*Henneken & A, 2011*; *Dorch, 2012*) and oceanography (*Sears, 2011*). As citations to papers for many disciplines are a the key metric by which impact of researchers is measured, this could mean a very important incentive to researchers for sharing their data. Moreover, there is a tendency to regard datasets as research output that can be used as a citeable reference or source in their own right (*Costello et al., 2013*; *Neumann & Brase, 2014*). For the field of oceanography it was found that datasets can be cited even more than most papers (*Belter, 2014*). This would mean that sharing datasets in the near future could have a direct positive influence on a researcher's scientific impact.

On the other side of the coin, a researcher who reuses a dataset that was shared can gain several advantages. Time is saved in not having to collect or produce the data, which can be put to use to produce more papers. Papers can be enhanced with a comparison or meta-analysis based on an extra dataset. If the added dataset merits publication in a higher impact journal, the paper could be cited more often. In more general terms, the scientific community can benefit from reuse of datasets. Sharing data enables open scientific inquiry, encourages diversity of analysis and opinion, promotes new research, facilitates the education of new researchers, enables novel applications to data not envisioned by the initial investigators, permits the creation of new datasets when data from multiple sources are combined, and provides a basis for new experiments (*Ascoli, 2007*; *Kim, 2013*; *Pitt & Tang, 2013*). It also is a way to prevent scientific fraud; with the dataset provided, one should be able to reproduce scientific results.

To summarize, data sharing implies costs and/or benefits for the individual researcher, but are of clear benefit to researchers that reuse the dataset, and to the scientific community as a whole. In this context, the problem of data sharing can be studied as a game theoretic problem. The strength of game theory lies in the methodology it provides for structuring and analysing problems of strategic choice. The players, their strategic options, the external factors of influence on those decisions, all have to be made explicit. With the model, we show how research data sharing fits the definition of a typical 'tragedy of the commons,' in which cooperating is the best strategy but cheating is the evolutionary stable strategy. In addition, we assess measures for altering costs and benefits with sharing and reuse and analyse how each measure would turn the balance towards *more* sharing and *more* benefits from sharing, benefitting the community, society and the individual researcher.

## METHODS

### A model for impact

We assume a community of researchers who publish papers. We consider two types of researchers: those sharing and not sharing research data associated with those papers. We make the simplifying assumption that the goal for both types of researchers is to perform well by making a significant contribution to science, i.e., to have a large impact on science. We assume that produced papers, $P_s$ for sharers and $P_{ns}$ for non sharers, create impact by getting cited a number of times $c$. We assume c is constant, which means we do not distinguish between low and highly cited papers. To increase their performance, researchers need to be efficient, i.e., they should try to minimize the time spent on producing a paper, so more papers can be produced within the same timeframe. Papers from which the dataset is shared gain an extra citation advantage, increasing the impact of that paper by a factor $b$. In our model we consider only papers with a dataset as a basis, i.e., no review or opinion papers. So, the performance of researchers is expressed as an impact rate, in terms of citations per year, i.e., the impact for sharing and non-sharing researchers is defined as

$$E_s = P_s \cdot c \cdot (1 + b) \quad E_{ns} = P_{ns} \cdot c. \tag{1}$$

From the above expressions it is clear that the difference in impact between sharing and not sharing researchers is to a large extent dependent on the number of publications $P$ per year. These publications can be expressed in terms of an average time to write a paper $T_s$ for sharers and $T_{ns}$ for not sharers.

$$P_s = \frac{1}{T_s} \quad P_{ns} = \frac{1}{T_{ns}}. \tag{2}$$

The time $T$ consists of several elements that we make explicit here. Each paper costs time $t_a$ to produce. Producing the associated dataset costs a certain time $t_d$. Sharing a dataset implies a time cost $t_c$. We do not distinguish between large and small efforts to prepare a dataset for sharing; all datasets take the same amount of time. We assume there is a certain probability $f$ to find an appropriate dataset for a paper from the pool of shared datasets $X$, in which case the time needed to produce a dataset $t_d$ is avoided. We do acknowledge that some time is needed for a good 'getting to know' the external dataset and to process it, resembled in the time cost $t_r$. We calculate the time to produce a paper by

$$T_s = t_a + \frac{t_d}{1 + f \cdot X} + \left( t_r - \frac{t_r}{1 + f \cdot X} \right) + t_c \quad T_{ns} = t_a + \frac{t_d}{1 + f \cdot X} + \left( t_r - \frac{t_r}{1 + f \cdot X} \right). \tag{3}$$

In these formulae, the pool of available datasets $X$ determines the value of the terms with $t_d$ and $t_r$. When $X$ is close to zero, the term with $t_d$ approaches $t_d$. This implies that everybody has to produce their own dataset with time cost $t_d$. In contrast, when $X$ is very large the term approaches zero, implying almost everyone can reuse a dataset and almost no time is spent in the community to produce datasets. Between these two extremes, the term first

rapidly declines with increasing $X$ and then ever more slowly approaches zero (see the plots in the last column in the figure in Appendix S2). This is under the assumption that at a small number of available datasets, adding datasets will have a profound influence on the reuse possibilities. If datasets are already superfluous, adding extra datasets will have less influence on the reuse rate. The term representing the effort to reuse a paper $t_r$ works opposite to the term representing $t_d$. When $X$ is close to zero, the term approaches zero, implying nobody spends time to prepare a set for reuse. When $X$ is very large the term approaches $t_r$; everyone spends this time because everyone has found a set for reuse.

While the pool of datasets $X$ determines the values of the terms with $t_d$ and $t_r$ and with that the number of shared datasets, at the same time the shared datasets accumulate in the pool of shared datasets $X$. To come to a specification of this pool size $X$ we formulate a differential equation for the pool size. A change in the pool of available, shared datasets $X$ depends on adding datasets belonging to papers $P_s$ from sharing researchers $Y_s$, minus the decay $q_x \cdot X$ of the datasets. Such a decay rate could be a result from a fixed storage time after which datasets would be disposed of or by a loss of data value, for instance by outdated techniques.

$$\frac{dX}{dt} = Y_s \cdot P_s - q_x \cdot X. \tag{4}$$

Using Formula (2) and (3) with the system at steady state i.e., $dX/dt = 0$, the pool size $X$ as function of the publication parameters and the size of the group of sharing researchers is given by

$$X = \frac{-\left(q_x(t_a + t_c + t_d) - Y_s f\right) + \sqrt{\left(q_x(t_a + t_c + t_d) - Y_s f\right)^2 - 4\left(q_x \cdot f(t_a + t_c + t_r)\right) \cdot (-Y_s)}}{2\left(q_x f(t_a + t_c + t_r)\right)} \tag{5}$$

(Formula (5) is derived in Appendix S1). So, for each parameter setting, we calculate $X$, and consequently, we calculate the impact in terms of citation rates $E_s$ and $E_{ns}$ with Formulae (1)–(3). Table 1 gives the default parameter settings that we use for our simulations.

## An individual based model

In addition to the model for impact we set up an individual based model to assess the impact for individual researchers depending on their personal publication rate, sharing and reuse habits, rather than to work with averages. We use the 'model for impact' as a basis for the calculations and then assign characteristics to individuals. First, a publication rate $P_r$ per researcher is assigned at random to individual researchers. $P_r$ is based on the distribution as seen in Fig. 1, fitted with the function

$$P_r = Y \cdot e^{-(t_a + t_d)}. \tag{6}$$

**Table 1 Parameters, variables, and their values.** Overview of parameters, variables, and their standard values used in the model. Grey rows indicate the parameters that are varied in the model to assess their influence (examples for real-world measures to change these are explained in Table 2).

| Parameter | Meaning | Value | Source | Unit |
|---|---|---|---|---|
| $t_a$ | Time-cost to produce a paper | 0.13 | Derived: $t_a + t_d$ amount to 121 days; leading to ∼3 papers a year (similar to the average in Fig. 1) | Year/paper |
| $t_d$ | Time-cost to produce a dataset | 0.2 | Derived: $t_a + t_d$ amount to 121 days; leading to ∼3 papers a year (similar to the average in Fig. 1) | Year/paper |
| $t_c$ | Time-cost to prepare a dataset for sharing | 0.1 | Estimated: 36.5 days | Year/paper |
| $t_r$ | Time-cost to prepare a dataset to reuse | 0.05 | Estimated: 18.25 days | Year/paper |
| $q_x$ | Decay rate of shared datasets | 0.1 | Derived: based on a storage time of 10 years | 1/year |
| $b$ | Citation benefit (sharing researcher) | 0 | Estimated: percent extra citations | Percent |
| $f$ | Probability to find an appropriate dataset | 0.00001 | Fitted | 1/dataset |
| $c$ | Citations per paper produced | 3.4 | Derived: approximate from 'baselines'; average citation rate by year three, Thompson Reuters | Citation/paper |

| State variables | Meaning | Value | | Unit |
|---|---|---|---|---|
| $E$ | Impact | See formula (1) | Calculated | Citation/year |
| $P$ | Number of papers | See formula (2) | Calculated | Paper/year |
| $T$ | Time for a publication | See formula (3) | Calculated | Year/paper |
| $X$ | Pool of shared datasets | See formula (5) | Calculated | Dataset |
| $Y$ | Number of researchers | 10000 | Defined | n.a. |

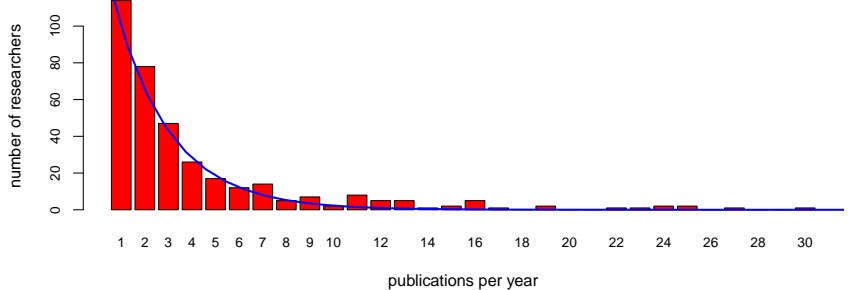

**Figure 1 Publication distribution.** The sampled (bars) and fitted (line) distribution of published papers per researcher in a given year, in this case 2013. For reasons of visualisation the distribution is shown up to thirty publications, whereas the sampling sporadically included more publications per researcher. The fitted line is used as the published papers' distribution for the simulated community.

As a next step we introduce parameters that have to do with sharing. The percentage of sharing researchers is a fixed parameter in this model. The researchers sharing type is assigned at random to individuals. The actual reuse of a dataset, based on the probability to find an appropriate dataset for a paper, is assigned at random to publications. The portion of papers $R$ for which an appropriate dataset for reuse is found is calculated as

$$R = 1 - \frac{1}{1 + f \cdot X}. \tag{7}$$

We now have a mix of individual researchers that share or do not share, find a dataset for reuse or not for any of their papers, and publish different number of papers in a year. Based on the parameters in Table 1 we assign costs and benefits with these traits. These factors determine the performance of researchers in terms of impact by citations.

To determine the publication rate distribution in Fig. 1, we sampled the bibliographic database Scopus. We selected the first four papers for each of the 26 subject areas in Scopus-indexed papers, published in 2013. If a paper appeared within the first four in more than one subject area, it was replaced by the next paper in that subject area. For each of the selected papers, we noted down all authors and checked how many papers each author (co-) authored in total in 2013. We came to 366 unique authors in our selected papers. Authors that were ambiguous, because they seemingly published many papers, were checked individually and excluded if it was a group of authors publishing under the same name with different affiliations between the papers. For the data, see *Pronk, Wiersma & van Weerden (2015)*. This distribution, based on our sampling, implies that most researchers publish one- and a few researchers publish many papers in a given year. We fitted an exponential distribution through the sampled population (Formula (6)). The average for the distribution is close to three papers per researcher in a given year.

### Simulations

For the R-scripts to generate the plots for all simulations, see *Pronk, Wiersma & van Weerden (2015)*.

We start with a set of simulations regarding performances per sharing type, with the model for impact. We calculate the impact for the two types of researchers over a range of sharing from zero to a hundred percent of all researchers. In addition to the default values (see Table 1), we change parameters to assess their influence on the publication rate and associated impact by citations for sharing and not sharing researchers. In Table 2 we list the parameters changed in the simulations and a score of the measures that would have these effects in a 'real world' scientific community (*Chan et al., 2014*).

To have a closer look on individual performance, we perform the same set of simulations with the individual based model. For each setting, we calculate the difference between the publication rate assigned in Formula (6) at no costs or benefits with sharing or reuse, and a new, calculated publication rate based on sharing and reuse traits per researcher under the assumption that half of the researchers share. So, again we change the parameters in Table 2 and assess their influence, as in the first simulation.

We end by zooming out to community performance with the model for impact. We calculate the average impact over all researchers in the community, now at more extreme settings of the citation benefit $b$ and in a second simulation at even higher cost $t_c$ for preparing a dataset for sharing. This is to provide a broader range of results. Citation benefit $b$ and the sharing rate are changed within their range in one hundred equal steps.

## RESULTS

Shown in Fig. 2 are the simulations with the model for impact (Formulae (1)–(5)). The simulation in (A) is at default parameter values (Table 1). In (B–F) we simulated

**Table 2  Changed parameters and associated measures.** Overview of considered parameters determining reuse and sharing habits of researchers, and possible measures to improve these in a realistic setting.

| Parameters investigated in the model | Possible associated measures to improve this |
|---|---|
| Time '$t_r$' spent to assess and include an external dataset | • Improve data quality, for instance by the use of data journals (*Costello et al., 2013*; *Atici et al., 2013*; *Gorgolewski, Margulies & Milham, 2013*), or peer review of datasets (i.e., a 'comment' field in data repositories).<br>• Offer techniques or tools for easy assessment of dataset quality i.e., (*Eijssen et al., 2013*), faster pre-processing or data cleaning (i.e., 'OpenRefine' or 'R statistical language'). |
| Chance '$f$' to find an external dataset | • Harvest databases through data portals to reduce 'scattering' of datasets.<br>• Standardization of metadata and documentation.<br>• Advanced community and project-specific databases.<br>• Library assistance in finding and using appropriate datasets. |
| Time '$t_c$' associated with sharing of research data | • Offer a good storing & sharing IT infrastructure.<br>• Assistance with good data management planning at the early stages of a research project. |
| Benefit in citation per paper '$b$' associated with sharing of research data | • Provide a permanent link between paper and dataset.<br>• Increase attribution to datasets by citation rules .<br>• Establish impact metrics for datasets. |
| Percentage of scientists sharing their research data | • Promote sharing by a top down policy from an institute, funder, or journal.<br>• Promote sharing bottom up by offering education on the benefits of sharing, to change researchers' mind set. |

measures to improve upon impact. There are two important observations. First, in all (but the last) subfigure of Fig. 2A–2E) the average impact of not sharing researchers exceeds that of sharing researchers irrespective of how many sharing researchers there are. This means that *not sharing* is the best option, at all percentages sharing researchers. In this scenario, it would be logical if all individual researchers would choose not to share and eventually end up getting the average impact by citations depicted at zero percent sharing. So we see here a classical example of the tragedy of the commons or prisoners dilemma phenomenon. What is important to note though is that the measures in (B) (C) (D) and (E) ascertain a key effect when compared to the default in (A). The average impact of sharing researchers at the highest percentage sharing researchers (straight horizontal light-grey line; stripes) is increasingly higher with the measures than the average impact for not sharing researchers at zero percentage sharing researchers (straight horizontal dark-grey line; dots–stripes). Should a policy enforce the sharing, or all would agree to cooperate and share, a higher gain is achieved than in the case that researchers would all choose not to share. This illustrates the conflicting interest for individual researchers, who are better off not sharing, while they would do better if all of them did share. Subfigure (F) of Fig. 2 shows the potential of the citation benefit with sharing. In the picture it is profitable to share at low sharing rates, and profitable not to share at high sharing rates, leading to a stable coexistence of sharing and not sharing researchers. This means that the community would exist of researchers from both strategies. Hypotheti-

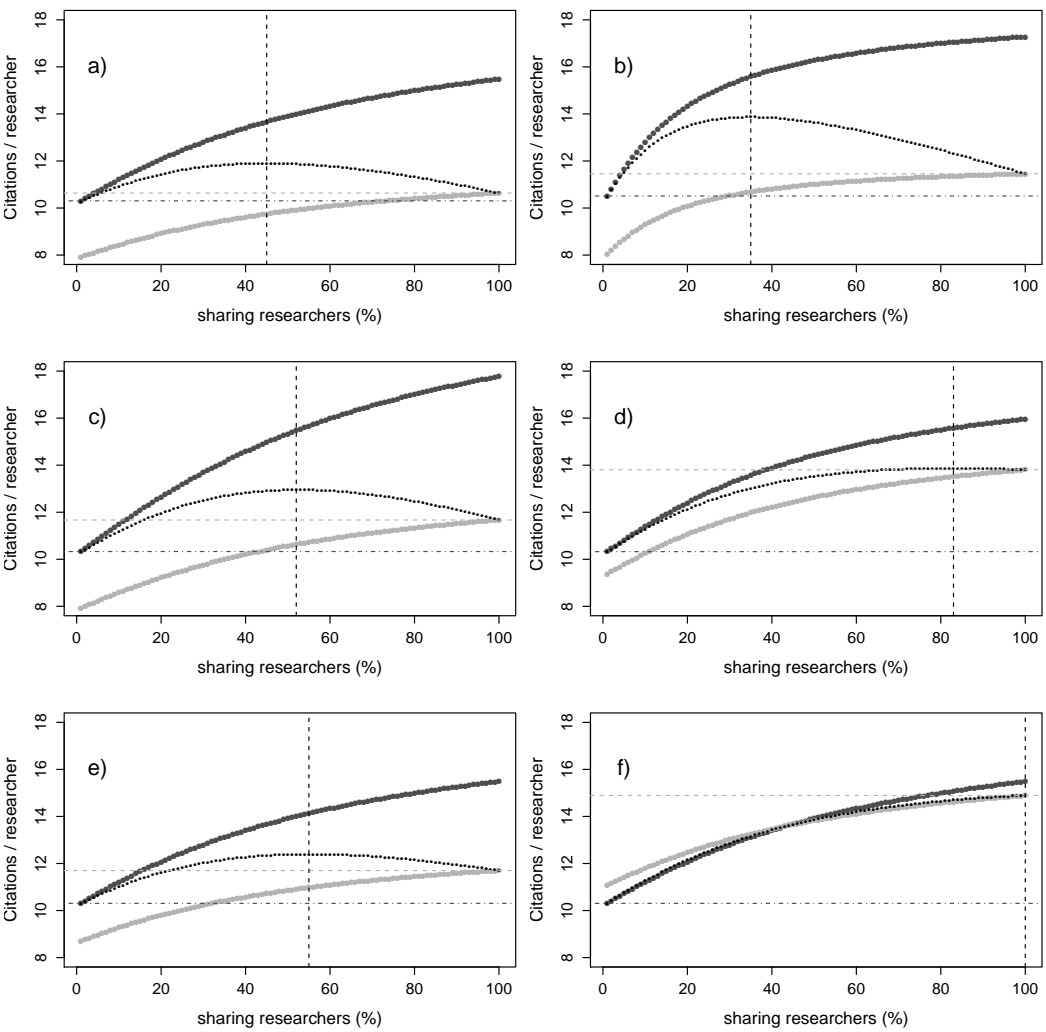

**Figure 2 Impact per sharing type.** Citations ('impact') per year for researchers sharing and not sharing, at different percentages of sharing researchers. The simulations are done at parameter settings (A) default (see Table 1), (B) default but with $f$ increased threefold (C) default but with $t_r$ decreased threefold (D) default but with $t_c$ decreased threefold (E) default but with $b$ set to 0.1 (F) default but with $b$ set to 0.4. The curved light-grey line depicts the impact of the sharing researchers . The curved dark-grey line depicts the impact of the not sharing researchers. The thin dotted curved black line is the averaged community impact. The straight black vertical dotted line depicts the percentage of sharing researchers at which community impact is maximized. The straight horizontal lines respectively depict the impact at zero percent researchers sharing (dark-grey line; dots-stripes) and hundred percent sharing researchers (light-grey line; stripes).

cally, should the citation benefit be even higher, the sharing strategy would outperform the not sharing strategy at all sharing percentages. Researchers would in this case choose to share even without measures to promote sharing, simply because it directly increases their impact.

Second, it can be noted that in some subfigures of Figs. 2A–2C and 2E the average citations are the highest at intermediate sharing. This means that if sharing increases further, it has a detrimental effect on average community impact. This is because the model is

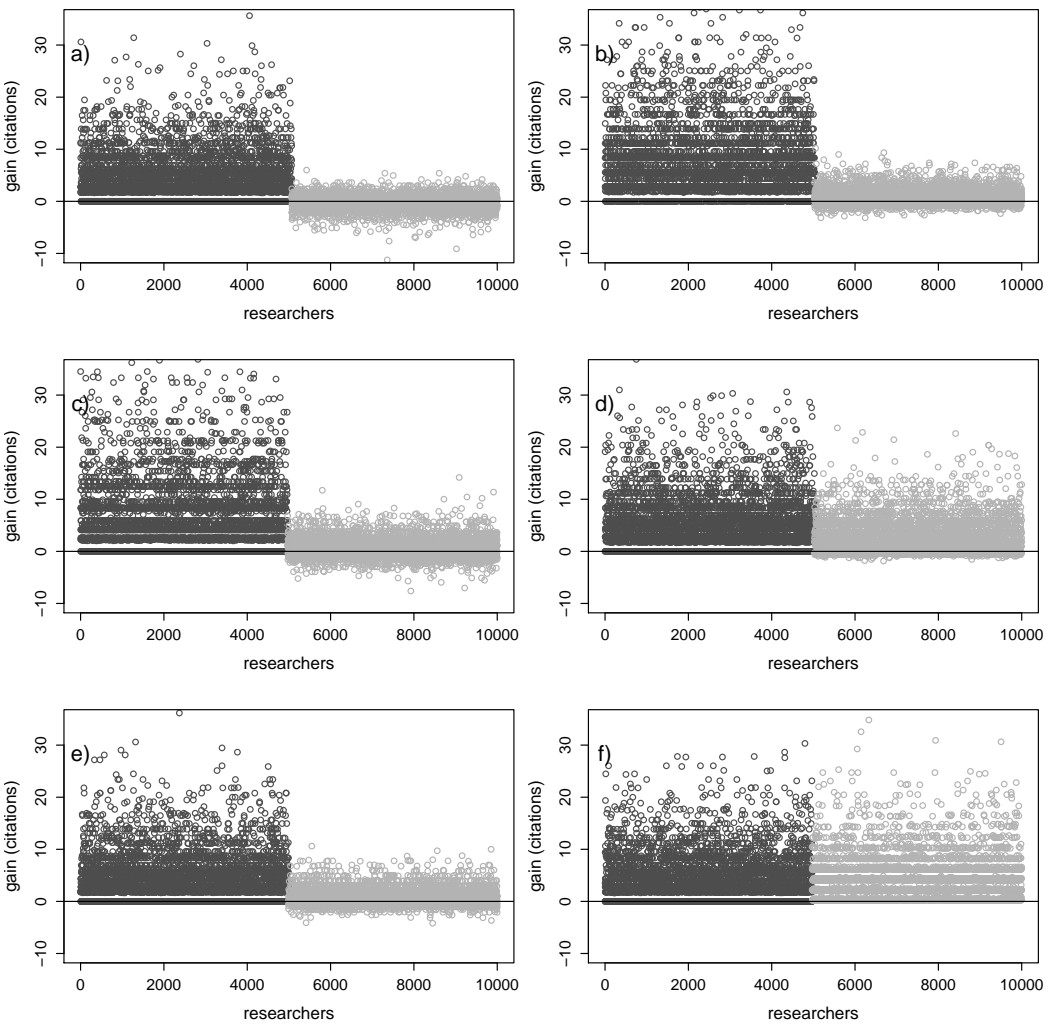

**Figure 3 Individual gains with sharing.** Gains from sharing in number of citations per individual researcher. These gains are calculated for the situation with fifty percent sharing researchers compared to the same situation without sharing researchers. For visualization purposes, the researchers are sorted according to sharing habitat: not sharing researchers (dark grey circles) to the left, sharing researchers (light grey circles) to the right. See the legend of Fig. 2 for parameter settings in all subfigures.

formulated in formula (3) in a way that total costs for sharing increase for the community as more researchers share, whereas total benefits cease to increase at high sharing rate. The extra datasets do not contribute much to the benefits, or in other words, the research community has become saturated with datasets. Compared to the average community citations, which are highest at intermediate sharing, for both sharing and not sharing researchers the highest impact by citations is at the point at which everyone is sharing.

Results from the individual based in Fig. 3 model show that the individual researchers have various gains depending on their publication rate, reuse, and dataset sharing habits. In (A) are the gains and losses in impact, at default parameter values (Table 1). In (B–F) we simulated measures to improve gains or limit losses. A possible desired effect of sharing of datasets would be that every individual researcher can benefit, sharing or not

sharing. It can be observed that in (Figs. 3A–3E) most of the sharing researchers have lower benefits or even costs compared to not sharing researchers. This logically is in line with the lower averages for sharing researchers in Fig. 2. Also, it can be noted in all subfigures of Fig. 3 that there are always sharing researchers that do not benefit from the availability of datasets by the reuse of datasets. These researchers were not (fully) able to compensate for the cost to share their data. It is notable that in (B) individual researchers are left with lower costs than in (C). This is because in (B) the probability of finding an appropriate dataset for reuse $f$ is set higher, compensating the sharing costs for many of the researchers. In (C) the time cost $t_r$ with reuse per paper is lower, benefitting only those few researchers that do find a reusable set. In (D) the lowering of the time cost $t_c$ for preparing a dataset for sharing improves the situation for *all* researchers compared to the default in (A), but still some researchers are not fully compensated. In (E) the introduction of the citation benefit $b$ does not help much to improve the benefits for sharing researchers. Only when in (F) a substantial citation benefit $b$ is introduced for sharing researchers, the costs associated with sharing are (more than) compensated for, for all sharing researchers.

When simulating community impact in Figs. 4A and 4B it can be seen that, as the benefits $b$ for sharing increase towards the right of the plot, the average community impact increasingly starts to rise with more sharing in both plots. Even the drop after the initial increase at increased sharing caused by the datasets saturation is eventually compensated for with the increase of the citation benefit with sharing. In subfigure (B) at the left side of the plot, without a citation benefit and with the very high cost for sharing $t_c$, there appears an alarming effect. At these parameter values the average impact becomes lower at high sharing than at no sharing at all. Policies increasing sharing would, if successful, in this case backfire and reduce scientific community impact.

## DISCUSSION

We analysed the effect of sharing and not sharing research data on scientific community impact. We found that there is a conflicting interest for individual researchers, who are *always* better off not sharing and omitting the sharing cost while they would have higher impact when sharing as a community. With our model we assessed some measures to improve the costs and benefits with sharing and reuse of data, to make most researchers profit from the sharing of datasets. We simulated policies to increase sharing, measures to stimulate reuse by reducing reuse costs or increasing discoverability of datasets, and measures to stimulate sharing by lowering costs associated with sharing or introducing a citation benefit with each shared dataset. These simulations concretize the notion in literature that improving spontaneous participation in sharing datasets will require lowering costs and/or increasing benefits for sharing (*Smith, 2009*; *Roche et al., 2014*) and values different measures to do so.

A policy is a straightforward measure to increase community impact simply by enforcing higher percentages of sharing researchers. Moreover, policies are pivotal for establishing acceptable data sharing practices and community-level standards. Such

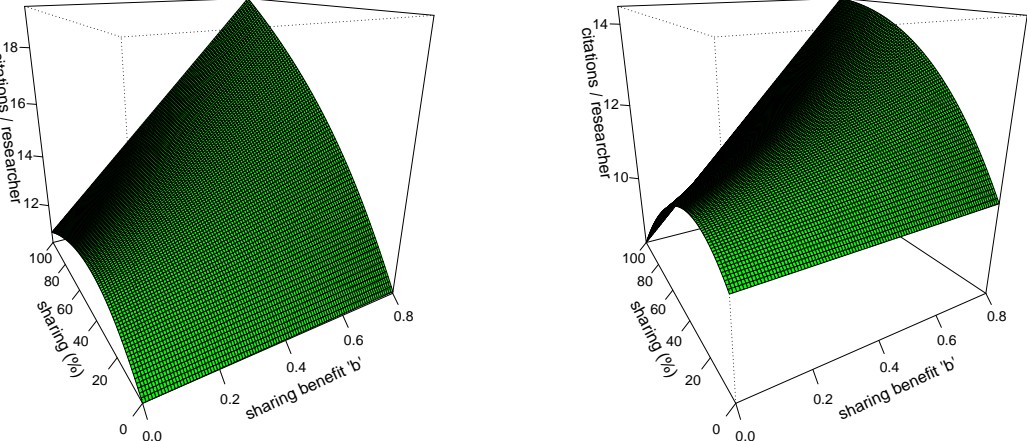

**Figure 4 Community impact.** Average community impact with varying percentage of sharing researchers and varying sharing benefit $b$. Figures are calculated at default parameter values (see Table 1) with the exception of $b$ which is varied, and for subplot (B) $t_c$, of which the value was set from 0.1 to 0.2. On the $z$-axis is the average community impact. On the x and y axes, respectively, increasing benefits $b$ for sharing from 0 to 0.8 (0 to 80% citation benefit with sharing) and increasing percentage of sharing researchers from 0 to 100%.

policies can be enforced on the level of institutions, funders, or journals. In the model these do increase community impact, as long as the community is not already saturated with datasets. In real life, at least for journals, policies have not been enough to convince researchers to actually make their dataset publicly available (*Wicherts et al., 2006*; *Savage & Vickers, 2009*; *Alsheikh-Ali et al., 2011*; *Wicherts & Bakker, 2012*; *Vines et al., 2013*). This could be exemplary for the reluctance of individual researchers to share datasets because of real, or perceived costs. The inequality in costs between sharing researchers and not sharing researchers remains with mandated sharing, and the researcher that does not share a dataset but does reuse a dataset will have the highest impact compared to all others. Of course there are many factors for researchers to decide to share data or not, but simply said this could predispose a researcher towards not sharing. The 'reuse-don't share' strategy is a true current sentiment towards using: according to a survey in 2011 of about 1,300 scientists, more than 80 percent said they would use other researchers' datasets but only few wanted to make their dataset available to others, for a variety of reasons (*Tenopir et al., 2011*; *Fecher et al., 2015*).

Stimulating reuse by reducing reuse costs or increasing discoverability of datasets in the model increases average community impact, though not equally for all individuals within the community. Only the researchers that actually reuse a dataset profit from these measures, and the costs for those who share, although partly compensated, still exist. Again, although helpful, the inequality in costs between sharing and not sharing researchers is not addressed with such measures.

A direct reduction of the time costs with sharing a dataset in our model improved the situation for all sharing researchers. Only a small inequality between sharing and not sharing researchers remained. The best solution is however to introduce a 'citation benefit' for papers with the dataset shared, to directly balance the costs of sharing indi-

viduals. The citation benefit in real life can not only come from increased citations to the paper (*Botstein, 2010*; *Sears, 2011*; *Dorch, 2012*; *Piwowar & Vision, 2013*) but also from citations to the shared dataset itself (*Costello et al., 2013*; *Belter, 2014*; *Neumann & Brase, 2014*). With a relatively high citation benefit, sharing datasets even becomes more profitable than not sharing, at any percentage of sharing researchers. Sharing then is not only optimal for maximizing community impact, but also for the individual researcher.

All in all, enhancement of the citation benefit would bring about better incentives to share datasets than imposing an obligation to share by funders, institutes or journals, or partly compensating for costs by enabling reuse. Better incentives arguably also lead to better sharing practices as researchers would strive to present their dataset as such that its reuse potential is optimal.

All models come with simplifications and assumptions. A central assumption of the model is the gain of scientific impact by citations to papers, and implicitly datasets. For some communities the concept of impact by citations is less applicable overall (*Krell, 2002*). These fall outside the scope of this model. It also should be noted that there are other ways to count scientific impact such as Altmetrics (*Roemer & Borchardt, 2012*). Additionally, we derived general phenomena for the scientific community, whereas (perceived) costs and benefits with sharing will differ between scientific communities (*Vickers, 2011*; *Tenopir et al., 2011*; *Kim, 2013*) and attitudes towards sharing can differ largely between disciplines (*Kirwan, 1997*; *Huang et al., 2012*; *Pitt & Tang, 2013*; *Anagnostou et al., 2013*). This means that the measures taken to make sharing worthwhile will have to differ in their focus in each scientific community (*Borgman, Wallis & Enyedy, 2007*; *Acord & Harley, 2013*). To apply the current model to any specific situation or community, parameter values for that community should be carefully determined and, where necessary, the model should be adjusted or expanded. Additional factors that may influence the outcome of this model and that could possibly be incorporated in community specific versions or future refinements of this model include: differences in quality of papers leading to differences in citation rates, heterogeneity in the costs of sharing (small and easy versus big and complicated datasets to document), heterogeneity in the contribution of a papers' dataset to the available pool of datasets, introducing and allowing for heterogeneity in search time for datasets, feedback between the number of times a dataset is reused and the citation benefit for that dataset. A focal point to assess in the current model would also be the pool of available datasets. What is the relation between available datasets and reuse rate for researchers, do these datasets overlap in content, will all new datasets contribute to science, does the pool become saturated, are all datasets reused, what is the decay rate of datasets in the pool for that specific community?

Lastly, it is clear that not all data can or should be made fully or immediately publicly available for a variety of practical reasons (e.g., lack of interest, sheer volume and lack of storage, cheap-to-recreate data, high time costs to prepare the data for reuse, the wish to publish later perhaps, patents pending, privacy sensitive data) (*Kim, 2013*; *Cronin, 2013*). With our simulations we show that if costs for sharing are too high relative to the benefits of reuse, in theory sharing policies to increase sharing could even backfire and

reduce scientific community impact. It should be carefully considered whether the alleged benefits of storage for the scientific community will outweigh the costs for each data type and set. For easily obtainable data such as the data underlying this paper, recreating it is probably cheaper than storing and interpreting the datasheet.

In conclusion, we performed a game-theoretic analysis to provide structure and to analyse problems of strategic data sharing. In the simulations there appeared a conflicting interest for individual researchers, who are *always* better off not sharing and omitting the sharing cost, while they are ultimately better off all sharing as a community. Although policies are indispensable and should be able to increase the rate of sharing researchers, and increased discoverability and dataset quality could partly compensate for costs, a better measure to promote sharing would be to lower the cost for sharing, or even turn it into a (citation-) benefit.

## ACKNOWLEDGEMENTS

We thank Dorinne Raaimakers, Jeroen Bosman, Jan Molendijk, Conny van Bezu from Utrecht University Library, Sebastiaan Wesseling from Toxicology, WUR and Mark van Oorschot from PBL, RIVM for their constructive ideas concerning the manuscript and initial concept. We thank two anonymous reviewers and Patricia Soranno, Gyuri Barabas, and Timothée Poisot for pointing out possibilities for improvement in previous versions of the manuscript.

### Funding
The authors received no funding for this work.

### Competing Interests
The authors declare there are no competing interests.

### Author Contributions
- Tessa E. Pronk conceived and designed the experiments, performed the experiments, analyzed the data, contributed reagents/materials/analysis tools, wrote the paper, prepared figures and/or tables, reviewed drafts of the paper.
- Paulien H. Wiersma performed the experiments, reviewed drafts of the paper.
- Anne van Weerden analyzed the data, reviewed drafts of the paper.
- Feike Schieving conceived and designed the experiments, analyzed the data, contributed reagents/materials/analysis tools, reviewed drafts of the paper.

### Data Availability
DataverseNL
http://hdl.handle.net/10411/20328 V4 [Version].

## Supplemental Information

Supplemental information for this article can be found online at http://dx.doi.org/10.7717/peerj.1242#supplemental-information.

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
