# Peer review of "A game theoretic analysis of research data sharing"

_PeerJ, doi:10.7717/peerj.1242_

## Round 0.1 · original submission · Minor Revisions

· Academic Editor

Minor Revisions

I am happy to see the resubmission has been obviously improved by the authors. I apologize for this (maybe) a little bit long review process, it took some time to ask for potential reviewers. Both reviewers suggested minor revisions, and gave their advices on the title of the manuscript and the methods part. I encourage you to make further revisions based on these advices. Personally, I am happy to receive two Minor Revisions from the reviewers, and prepare to accept the manuscript if you will successfully address these issues.

·

Basic reporting

The paper is generally clear, and the model is well explained. My only issue with the reporting would be the title. I read this paper as a preprint, and passed it along to colleagues who thought it would be a "game", as opposed to "game theory". I would suggest something similar to "The game theory of research data sharing", or any other title that emphasizes the game-theoretical underpinning of the research.

Experimental design

No comments -- the design is robust, and the replies to previous reviewers claims are good.

The code to reproduce the figures is given, and citeable, which is a high standard of quality of speaks highly of the authors dedication to data sharing.

Validity of the findings

The conclusions are expressed in plain language that will be accessible to a broad audience, and (although I disagree with some of the interpretation, see "General comments") are well supported by the experiments.

Additional comments

I have a single general comment. Reading through the paper, and the abstract in particular, I had the feeling that the authors were minimizing the potential benefits of strong policies (i.e. mandated data sharing/archival). I'm not sure this is true and/or relevant for a couple of different reasons.

1. By and large, such policies are *already* implemented by journals.
2. The results hold true assuming that all scientists behave rationally with regard to data sharing (which is not true), and especially accept that re-using existing data is an alternative to sampling.

I wish these limitations could be discussed in a revision. The last thing we need is less community-level standards for acceptable data sharing practices, and these will take the shape of policies.

·

Basic reporting

The paper is mostly well-written, except the Methods section. I believe this will be very easy to fix though, so it is not a serious problem at all. See my detailed comments below.

Experimental design

The paper contains no experiments. The model study is good but can be improved; see my comments below.

Validity of the findings

As far as I understand, the findings are basically sound; see my comments below for possible improvements.

Additional comments

In this manuscript the Authors investigate the individual and
community-wide costs and benefits of sharing scientific data with the
scientific community at large. It is assumed that, once a dataset is
published, it can be reused, increasing the productivity of scientists
in general, but preparing the dataset for publication has a time cost
paid by individual researchers. The Authors set up a game-theoretical
model to investigate the costs and benefits associated with data
sharing. They consider both a simpler, deterministic version of their
model and a more complicated individual-based simulation of the same
underlying ideas.

Please see my comments on the manuscript below. I begin with broad
issues and end with more specific coments.


BROAD ISSUES

Though the manuscript is generally well-written and easy to follow,
this is not true of the Methods, which could be streamlined somewhat
(see some of my further comments below, trying to help with this
issue). My most important concern is that of Equation 3. There are two
problems here. First, the equations are dimensionally incorrect ([T] =
years/paper, but [t] = years), which needs to be fixed. Second, unless
I misunderstood something, the equation contains the same search times
and "get-to-know" times whether researchers share data or not. But if
researchers prepare their own dataset, they 1) may not have had to
search for one to begin with, making the searching time possibly zero;
and 2) definitely did not have to spend any time on getting to know
the dataset they themselves created. I am wondering if this could be
incorporated into the model to see if it changes any of the results
substantially.


SPECIFIC COMMENTS

- l.79: "The fixity of c..." -> "We assume c is constant, which means
we do not distinguish..."

- l.81: "spend" -> "spent"

- l.83: "and this adds to the impact of that paper with b" ->
"increasing the impactof that paper by a factor b"

- Equations 2, 3, 5, and 7: the fraction lines are not showing up
properly.

- l.96-98: If I understand correctly, f is not the raw probability,
but the per dataset probability of finding an appropriate dataset;
otherwise the denominators in Equation 3 would be incorrect. Pease
also fix this issue in Table 2.

- l.104: "On the contrary" -> "In contrast"

- l.123-124: I think it would be notationally easier on readers to
write the standard dX / dt instead of d_t X for the time derivative.

- l.135-137: Please explain before writing down the equation that this
equation is supposed to be the fitting curve (am I right?) for the
observed author-publication distribution. E.g.: "A publication rate
P_r per researcher is assigned at random to individual
researchers. P_r is based on the distribution of seen on Figure 1,
fitted with the function [include Equation 6 here]."

- l.137, 158: Relating to the same issue: are there other curves that
fit well, e.g., lognormal, or power-law? I think it would be worth
trying to fit these alternative curves. If they also fit well, it
would be interesting to see how and whether the results depended on
this choice. If they do not fit well, that's another argument in
favor of the exponential curve. Alternatively, the Authors could
simply use the empirically observed curve as their probability
distribution.

---

## Round 0.2 · accepted · Accept

· Academic Editor

Accept

The authors have clarified and answered the questions from reviewers. I am happy to accept this revised version now.